# COVID-19 vaccine effectiveness in preventing severe outcomes and assessing the impact of prior SARS-CoV-2 infection among hospitalized adults in Albania, July 2022-July 2023

Jonilda Sulo[1,2]*, Kujtim Mersini[3,4], Elona Kureta[1], Iris Hatibi[1], Najada Como[5,6], Esmeralda Meta[5], Migena Qato[5], Eugena Tomini[6], Valbona Zefi[1,3], Juljana Nanaj[1,3], Silvia Bino[1,3,6]

1 Institute of Public Health, Control of Infectious Diseases Department, Tirana, Albania, 2 Mediterranean and Black Sea Programme in Intervention Epidemiology Training (MediPIET), European Centre for Disease Prevention and Control (ECDC), Stockholm, Sweden, 3 Southeast European Center for Surveillance and Control of Infectious Disease, Tirana, Albania, 4 Agricultural University of Tirana, Faculty of Veterinary Medicine, Tirana, Albania, 5 University Medical Center of Tirana "Mother Teresa", Tirana, Albania, 6 University of Medicine, Tirana, Albania

* jonildasulo@gmail.com

## Abstract

The COVID-19 pandemic has significantly affected Albania, with over 319959 cases and 3625 deaths by December 2023. Despite a national vaccination campaign initiated in January 2021, only 44.1% of the eligible population had completed the full vaccination regimen by late 2023. In this study, we aim to estimate the effectiveness of COVID-19 vaccines in preventing severe outcomes and in assessing association of prior SARS-CoV-2 infection among adults hospitalized with COVID-19 in Albania. A test-negative case-control study was conducted using SARI sentinel surveillance data from July 2022 to July 2023. The study included 1858 hospitalized SARI patients aged 18 and older, with 410 testing positive for SARS-CoV-2 (cases) and 1448 testing negative (controls). Vaccine effectiveness was calculated using logistic regression, adjusting for age, sex, comorbidities, and hospital stay duration. The overall adjusted vaccine effectiveness (VE) for preventing hospitalization was 30.1% for those fully vaccinated with two doses and 31.4% for those with a booster. VE against severe outcomes was 30.6%, increasing to 37.8% after a booster. The highest VE was observed in patients aged 80 and above, reaching 52.5% after a booster. Prior SARS-CoV-2 infection combined with vaccination was associated with higher VE to 76.2%. However, many VE estimates had wide confidence intervals overlapping zero, indicating considerable uncertainty and lack of statistical significance in some subgroup results. This study provides real-world evidence of an association between COVID-19 vaccination, particularly with booster doses, and lower odds of severe outcomes. VE was highest among elderly patients and in those with prior

**Data availability statement:** The data underlying this study originate from official national surveillance information systems held by the Institute of Public Health (IPH), Tirana, Albania. Due to the sensitive nature of these data and legal restrictions under national data protection regulations, the data are not publicly available and cannot be deposited in a public repository. Researchers who meet the criteria for access to confidential data may request access from the IPH, subject to approval and applicable data protection requirements, by contacting: ishp@shendetesia.gov.al.

**Funding:** The authors received no specific funding for this work.

**Competing interests:** The authors have declared that no competing interests exist.

SARS-CoV-2 infection, emphasizing the importance of booster campaigns in vulnerable populations.

---

## Introduction

The COVID-19 pandemic, caused by the SARS-CoV-2 virus, has swept the globe since it first emerged in late 2019. So far, it has led to a significant number of over 700 million confirmed cases and claimed the lives of more than 6 million individuals globally [1].

Albania, an upper-middle-income country with a population of approximately 2.8 million, has been substantially affected by the pandemic, reporting over 319959 confirmed COVID-19 cases and 3625 deaths by December 2023 [2,3]. This corresponds to approximately 11.4% of the total population, although confirmed case counts are likely to underestimate the true cumulative burden of infection because they do not capture undiagnosed, asymptomatic, or untested cases The country experienced multiple epidemic waves, initially driven by the Alpha and Delta variants during 2020–2021, followed by a marked surge in early 2022 associated with the emergence of the Omicron variant. Seroprevalence studies conducted by mid-2022 indicated that more than 90% of the Albanian population had developed antibodies against SARS-CoV-2, reflecting extensive viral transmission alongside moderate levels of vaccine-induced immunity [4]. With continued Omicron transmission, the proportion of the population with prior infection or hybrid immunity during 2022–2023 was likely higher than indicated by confirmed case counts alone.

In response to the pandemic, Albania launched a national COVID-19 vaccination campaign in January 2021, in accordance with recommendations from the World Health Organization Strategic Advisory Group of Experts on Immunization (WHO SAGE) and the European Technical Advisory Group [5]. Initial vaccination efforts prioritized health-care and social workers, residents of long-term care facilities, and educators, before expanding to the general population, including children aged 5 years and older, and the administration of booster doses for high-risk groups. As of December 2023, approximately 44.1% of the eligible population had completed the primary two-dose vaccination series, 16.5% had received at least one booster dose, and 46.1% had received at least one vaccine dose. Vaccines administered in Albania included Comirnaty (BNT162b2,Pfizer-BioNTech), ChAdOx1-S (AstraZeneca), Gam-COVID-Vac (Sputnik V), and CoronaVac (Sinovac Biotech) [6].

The effectiveness of COVID-19 vaccines in preventing infection and reducing severe cases has been extensively studied. In Europe, between December 2021 and July 2022, vaccine effectiveness (VE) against hospitalization due to SARS-CoV-2 was 43% for full vaccination, increasing to 59% with an mRNA booster, and peaking at 85% when the booster was received 14–59 days prior to infection [7]. During the Alpha variant period, VE was 85% with the complete Comirnaty series, but it fell to 54% during the Delta period, with a booster raising it to 91% [8]. Despite these efforts, data on VE in Albania remain scarce, particularly concerning severe outcomes and the impact of prior SARS-CoV-2 infection.

Drawing from the experience of the H1N1 pandemic in 2009, many WHO Member States in Europe established sentinel surveillance for severe acute respiratory infections (SARI), encompassing influenza. This surveillance system, involving systematic data collection, serves to monitor disease impact and severity, identify viruses causing severe symptoms, and establish risk factors for severe course of illness. Following WHO's guidance, countries with existing hospital-based sentinel influenza surveillance systems were urged to adapt them to monitor severe SARS-CoV-2 cases in addition to influenza, collecting data for measuring COVID-19 vaccine effectiveness (VE) when feasible [9].

Building upon the success of using SARI surveillance for real-life influenza vaccine effectiveness evaluation, this observational study in Albania leverages the existing hospital-based surveillance system to estimate the effectiveness of COVID-19 vaccines in preventing severe SARI cases with laboratory-confirmed SARS-CoV-2 infection.

In this study, we aim to estimate the effectiveness of COVID-19 vaccines in preventing severe outcomes and in assessing the association of prior SARS-CoV-2 infection among adults hospitalized with COVID-19 in Albania.

## Materials and methods

### Population and study design

SARI surveillance was established in Albania in 2009, primarily focusing on influenza and other respiratory pathogens. However, since 2022, the SARI surveillance system has undergone significant extensions. It is now integrated into an electronic infectious disease surveillance system (SISI), allowing identification and comprehensive testing of SARI samples for influenza, SARS-CoV-2, and other relevant pathogens. This enhanced integration provides a more robust understanding of respiratory infections in the country.

In Albania, there are eight sentinel sites for SARI surveillance located across seven regions as well as two national hospitals in capital city are enrolled in this system. These locations were chosen to provide a representative geographical sample covering 72% of the country's population.

We conducted a test-negative case control study among SARI patients over 18 years old who meet WHO case definition [8] and were hospitalized at one of the eight SARI sentinel sites from July 2022 to July 2023. This design involved analyzing data on vaccinated and unvaccinated patients based on the "test-negative design" (TND), where cases are SARI patients who tested positive for SARS-CoV-2, and controls are SARI patients, tested negative for SARS-CoV-2. Due to limited viral typing capacities, COVID-19 variants were not included in this study design. Variant-specific data were not available at the individual case level because genomic typing was not performed systematically for all SARS-CoV-2–positive SARI patients. Therefore, all laboratory-confirmed SARS-CoV-2 cases were analysed together as COVID-19–positive cases without stratification by variant or sub-variant. During the study period, the dominant circulating variant was omicron.

### Data collection

The SARI surveillance study participants were swabbed and tested for COVID-19 using real-time reverse transcription–polymerase chain reaction (RT-PCR) tests. Respiratory specimens were taken from all SARI patients within 48 hours of hospital admission, and PCR testing for COVID-19 and influenza was carried out at the Institute of Public Health (IPH). SARI cases were identified across a range of hospital wards within sentinel sites, including the Intensive Unit Care (ICU)/critical care unit, general/internal medicine, pediatric medicine, infectious disease ward, respiratory disease ward, and the maternity ward within the regional hospital. Required information on study participants includes demographic information, clinical and treatment details, and vaccination status for both COVID-19 and influenza infections. All the data collected were entered into the SISI system, while COVID-19 and influenza vaccination details were gathered from the electronic immunization system. Data were retrospectively retrieved from both systems for research purposes on 01.01.2024. For this analysis, authors had access only to de-identified datasets, and no directly identifiable personal information was available to the authors during or after data extraction.

## Inclusion and exclusion criteria

SARI patients aged ≥18 years with an available SARS-CoV-2 RT-PCR result and documented COVID-19 vaccination status were eligible for inclusion. As shown in Fig 1, of 3348 hospitalized SARI patients tested for SARS-CoV-2 between 1 July 2022 and 1 July 2023, 1490 were excluded: 1031 were aged <18 years; 13 did not meet the WHO SARI case definition; 35 had missing vaccine dose dates; 108 were swabbed >7 days after hospitalization; 54 were swabbed >10 days after symptom onset; 215 were influenza-positive [10]; and 34 had an unknown COVID-19 test result. The final analytic cohort therefore included 1858 eligible patients aged ≥18 years, comprising 410 SARS-CoV-2–positive cases and 1448 SARS-CoV-2–negative controls (Fig 1).

## Definitions

Patients meeting the WHO case definition [11] for SARI and who were eligible according to the inclusion-exclusion criteria were enrolled in the study. The study participants were divided into two groups – cases and controls. The cases were defined as confirmed COVID-19 cases, identified as patients admitted to the hospital with SARI symptoms and having a positive SARS-CoV-2 PCR test result obtained within 48 hours of admission or documented within 14 days prior to admission. The control group comprised SARI patients testing negative for SARS-CoV-2, within 48 hours, and were not tested positively 14 days prior to admission.

Patients were considered fully vaccinated with a primary series if they had received two doses of a COVID-19 vaccine at least 14 days before SARI symptom onset. Those who had received both doses followed by a booster dose at least 14 days before symptom onset were considered fully vaccinated with a booster [12]. Unvaccinated patients included those who had not received any COVID-19 vaccine or had received only one dose at least 14 days before symptom onset.

Comorbidities were defined as the presence of one or more chronic conditions documented in hospital medical records at the time of admission, including cardiovascular disease, immunocompromised conditions, diabetes, liver disease, neurological conditions, chronic kidney disease, chronic lung disease, malignant neoplasms, asthma bronchialis, and obesity.

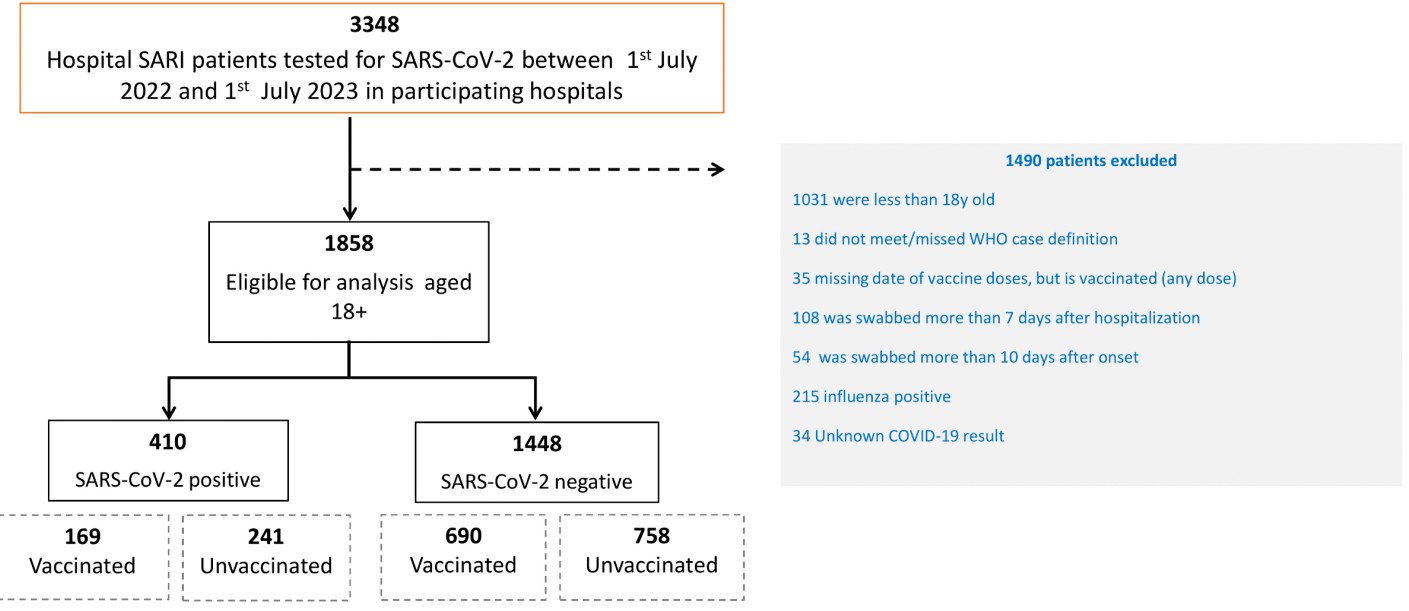

**Fig 1. Flowchart for SARI patient inclusion in the VE study, SARI sentinel surveillance, 1st July 2022- 1st July 2023.**

Individuals with a previous infection were defined as any eligible individual who tested positive for SARS-COV-2 by RT-PCR from January 2020 and up to 14 days before the current positive test.

Severe outcomes were defined as fatal outcome of patients or having one of the following requirements: the need for admission to the intensive care unit (ICU) for supplemental oxygen or mechanical ventilation.

## Statistical analysis

We summarized the demographic and clinical characteristics of both cases and test-negative controls, as well as the vaccinated and unvaccinated SARI patients. Discrete variables were expressed as frequencies and proportions, while continuous variables were summarized using medians and interquartile ranges (IQR). Comparisons of baseline characteristics between groups were made using Fisher's exact test for categorical variables and the Wilcoxon rank-sum test for continuous variables. Vaccine effectiveness was estimated for patients with a confirmed prior infection, severe outcomes, and across different age groups. For the age group analysis, patients were stratified into categories 18–59, 60–79, and ≥80 years to derive age-specific VE estimates. Each subgroup analysis was performed on a dedicated cohort restricted to the population of interest. Within each subgroup-specific cohort, the test-negative design was applied, designating SARS-CoV-2 positive patients as cases and negative patients as controls. The models were structured using multivariable logistic regression, where the SARS-CoV-2 test result served as the dependent variable and vaccination as the primary independent variable. Crude VE was calculated as VE = (1−OR) × 100%, with 95% confidence intervals (CI) provided for precision. Adjusted VE estimates were obtained by adjusting for age, sex, presence of at least one chronic condition (i.e., hypertension, heart disease, chronic respiratory disease, diabetes, liver disease, renal disease, immunodeficiency), and duration of hospital stay as a proxy for unmeasured confounding. All analyses were performed using R Statistical Software (v4.2.1; R Core Team 2021).

## Ethics statement

Based on the Albanian Law on Communicable Diseases No. 15/2016, which permits studies related to communicable disease surveillance to be conducted using existing surveillance systems, the IPH determined that the activities described in the study protocol are limited to public health surveillance and meet the criteria for routine surveillance as defined in this law. The data used in this analysis are derived from national surveillance systems that are established and regulated by law. In accordance with this legal framework, obtaining informed consent from patients is not required for public health surveillance activities. For the present analysis, only anonymized surveillance data were used.

## Results

From July 1, 2022, to July 1, 2023, 3348 hospital SARI patients were tested for SARS-CoV-2. Among them, 1490 participants were excluded due to not meeting the inclusion criteria. The final cohort included 1858 SARI patients aged 18 years and older.

Within this cohort, 410 (22.1%) tested positive for SARS-CoV-2 (cases), while 1448 (77.9%) tested negative (controls) (Fig 2).

The median age of the total study population is 66 years, with controls being younger (median age 64) and cases older (median age 71). The distribution by age group shows that 55.6% of cases are over 70 years old, compared to 34.1% of controls. There was an almost equal distribution between males and females in both groups.

Chronic conditions were present in 46% of both cases and controls, with notable differences in diabetes (21.7% in cases vs. 13.9% in controls) and neurological disease (10% in cases vs. 5.5% in controls). Conditions like kidney disease, lung disease, and cancer were more prevalent among cases.

In this study, 14.9% of the total study population had a history of a previous positive SARS-CoV-2 test, with a higher proportion among controls (16.1%) compared to cases (10.5%).

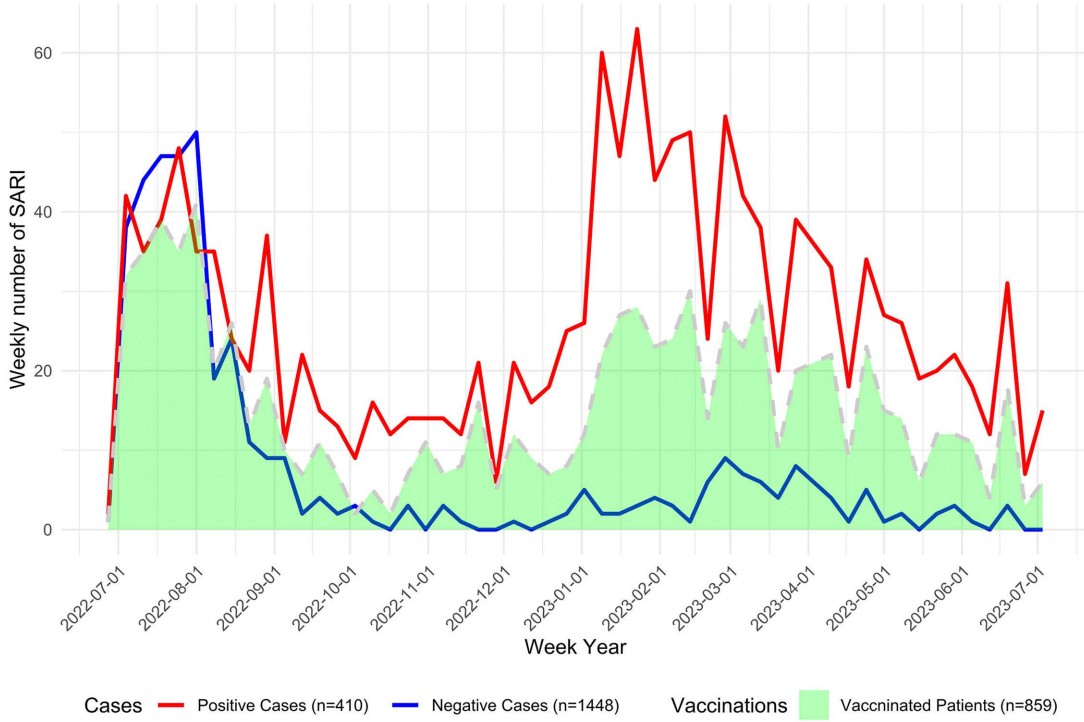

**Fig 2. Number of SARI COVID-19 positive (cases) and COVID-19 negative (controls) cases by week of result and number of SARI vaccinated by week of complete vaccination, Albania SARI sentinel surveillance, 1st July 2022- 1st July 2023.**

ICU admission was more frequent in controls (5.6%) than in cases (2.7%). Mechanical ventilation was used at similar rates for both cases (3.7%) and controls (3.3%), while CPAP usage was comparable between the two groups. Oxygen supplementation was more common in controls (21.8%) than in cases (11.2%), and the death rate was significantly higher among cases (15.6%) compared to controls (3.2%).

Regarding vaccination status, 58.8% of cases were unvaccinated, compared to 52.3% of controls. The percentage of those fully vaccinated with a primary course was higher among controls (27.4%) than cases (21.7%) while the percentage of booster vaccinated is slightly higher in control group 20.2% compared to the cases with 19.5%.

Hospital stay durations differ significantly between cases and controls, with controls more frequently having longer stays (p < 0.001) (Table 1).

We estimated VE against severe COVID-19 outcomes, prior infection, and across different age groups. The overall adjusted vaccine effectiveness in fully vaccinated with primary course cases is 30.1% (95% CI: 7.9 to 47.3), while for those who received a booster, it is 31.4% (95% CI: 8.2 to 49.2).

The VE among cases with severe outcomes was 30.6% (95% CI: -40.0 to 67.9) for those fully vaccinated with a primary course and37.8% (95% CI: -23.2 to 70.5) for those who received both a primary course and a booster shot. Vaccine effectiveness among patients with previous history of SARS COV-2 infection was 7.5% (95% CI: -101.8 to 58) for those fully vaccinated with a primary course and 76.2% (95% CI: 36.2 to 92.5) for those who received both a primary course and a booster shot.

The VE results among SARI patients reveal notable variations across different age groups and vaccination statuses. Among patients aged 18–59, the VE for two doses stands at 35.5% (95% CI:-7.01%-62.1%). For booster doses in this age group, the adjusted VE increases to 42.8%, with a 95% CI ranging from -24.4% to 77.1%. In the 60–79 age group,

**Table 1. Characteristics of SARI cases (n = 410) and controls (1448) recruited for the VE study, Albania SARI sentinel surveillance, 1st July 2022- 1st July 2023.**

| Characteristics | Value | Total (1858) | | Controls (n = 1448) | | Cases(n = 410) | | |
|---|---|---|---|---|---|---|---|---|
| Age | Median (IQR) | 66 | (53-74) | 64 | (51-73) | 71 | (62-79) | |
| | | n | % | n | % | n | % | p |
| Age groups | 18-29 | 154 | 8.3 | 138 | 9.5 | 16 | 3.9 | <0.001 |
| | 30-39 | 113 | 6.1 | 94 | 6.5 | 19 | 4.6 | |
| | 40-49 | 138 | 7.4 | 119 | 8.2 | 19 | 4.6 | |
| | 50-59 | 240 | 12.9 | 207 | 14.3 | 33 | 8.1 | |
| | 60-69 | 492 | 26.5 | 397 | 27.4 | 95 | 23.2 | |
| | >70 | 721 | 38.8 | 493 | 34.1 | 228 | 55.6 | |
| Sex | Male | 994 | 53 | 762 | 53 | 232 | 57 | 0.2 |
| | Female | 864 | 47 | 686 | 47 | 178 | 43 | |
| Pregnant | | 10 | 0.5 | 6 | 0.4 | 4 | 1 | 0.2 |
| Presence of chronic condition (one or more) | | 855 | 46.0 | 665 | 45.9 | 190 | 46.3 | >0.9 |
| | Cardiovascular disease | 678 | 36.5 | 536 | 37.0 | 142 | 34.6 | 0.4 |
| | Immunodeficiency | 10 | 0.5 | 10 | 0.7 | 0 | 0.0 | 0.13 |
| | Diabetes | 290 | 15.6 | 201 | 13.9 | 89 | 21.7 | <0.001 |
| | Liver Disease | 13 | 0.7 | 10 | 0.7 | 3 | 0.7 | >0.9 |
| | Neurological Disease | 120 | 6.5 | 79 | 5.5 | 41 | 10.0 | 0.002 |
| | Kidney Disease | 66 | 3.6 | 41 | 2.8 | 25 | 6.1 | 0.004 |
| | Lung Disease | 86 | 4.6 | 57 | 3.9 | 29 | 7.1 | 0.011 |
| | Cancer | 7 | 0.4 | 0 | 0.0 | 7 | 1.7 | <0.001 |
| | Asthma_bronchialis | 34 | 1.8 | 33 | 2.3 | 1 | 0.2 | 0.003 |
| | Obesity | 7 | 0.4 | 6 | 0.4 | 1 | 0.2 | >0.9 |
| History of a previous positive SARS-CoV-2 test | | 276 | 14.9 | 233 | 16.1 | 43 | 10.5 | <0.001 |
| Severity outcome | | | | | | | | |
| | ICU admission | 92 | 5.0 | 81 | 5.6 | 11 | 2.7 | 0.014 |
| | Mechanical Ventilation | 63 | 3.4 | 48 | 3.3 | 15 | 3.7 | 0.8 |
| | CPaP | 28 | 1.5 | 21 | 1.5 | 7 | 1.7 | 0.7 |
| | O2 | 362 | 19.5 | 316 | 21.8 | 46 | 11.2 | <0.001 |
| | Death | 110 | 5.9 | 46 | 3.2 | 64 | 15.6 | <0.001 |
| COVID-19 vaccination status | | | | | | | | 0.038 |
| | Fully vaccinated with a primary course | 486 | 26.1 | 397 | 27.4 | 89 | 21.7 | |
| | Fully vaccinated with a primary course plus booster | 373 | 20.1 | 293 | 20.2 | 80 | 19.5 | |
| | Unvaccinated | 999 | 53.8 | 758 | 52.3 | 241 | 58.8 | |
| Length of hospital stay | | | | | | | | <0.001 |
| | < 5 days | 662 | 35.6 | 448 | 30.9 | 214 | 52.2 | |
| | 5-15 days | 869 | 46.8 | 705 | 48.7 | 164 | 40.0 | |
| | 15-30 days | 201 | 10.8 | 182 | 12.6 | 19 | 4.6 | |
| | > 30 days | 126 | 6.8 | 113 | 7.8 | 13 | 3.2 | |

both two doses and booster doses exhibit lower VE percentages. For two doses, the VE is 15.5% with a 95% CI ranging from -22.0% to 42.1%, and for booster doses, the VE is 15.0% with a 95% CI ranging from -21.6% to 41.0%.

Patients over 80 years old demonstrate higher VE percentages. For two doses, the VE is 47.8% with a 95% CI ranging from -15.3% to 77.6%, and for booster doses, the VE is 52.5%, with a 95% CI ranging from 1.8% to 78.1% (Table 2).

**Table 2. Effectiveness of complete vaccination against COVID-19 hospitalization among SARI patients, Albania SARI sentinel surveillance, 1st July 2022- 1st July 2023.**

| | Cases | Controls | Crude VE | 95% CI | Adjusted VE | 95% CI |
|---|---|---|---|---|---|---|
| Overall | | | | | | |
| Unvaccinated | 241 | 758 | | | | *reference* |
| Fully vaccinated with a primary course | 89 | 398 | 29.5 | 7.8-46.5 | 30.1 | 7.9-47.3 |
| Fully vaccinated with a primary course plus booster | 80 | 293 | 14.1 | -14-35.8 | 31.4 | 8.2-49.2 |
| Severe Outcome | | | | | | |
| Unvaccinated | 44 | 188 | | | | *reference* |
| Fully vaccinated with a primary course | 11 | 68 | 30.8 | -37.2-67.7 | 30.6 | -40.0-67.9 |
| Fully vaccinated with a primary course plus booster | 12 | 75 | 31.6 | -32.9-67.1 | 37.8 | -23.2-70.5 |
| Previous Infection | | | | | | |
| Unvaccinated | 25 | 108 | | | | *reference* |
| Fully vaccinated with a primary course | 13 | 62 | 9.4 | -87.1-57.7 | 7.5 | -101.8-58.0 |
| Fully vaccinated with a primary course plus booster | 5 | 63 | 65.7 | 12.7-88.8 | 76.2 | 36.2-92.5 |
| Age- Group | | | | | | |
| 18–59 years | | | | | | |
| Unvaccinated | 56 | 307 | | | | *reference* |
| Fully vaccinated with a primary course | 24 | 188 | 30 | -15.4-58.6 | 35.5 | -7.01-62.1 |
| Fully vaccinated with a primary course plus booster | 7 | 63 | 39.1 | -31.7-75.6 | 42.8 | -24.4-77.1 |
| 60–79 years | | | | | | |
| Unvaccinated | 120 | 357 | | | | *reference* |
| Fully vaccinated with a primary course | 54 | 183 | 12.2 | -26.2-39.5 | 15.5 | -22.0- 42.0 |
| Fully vaccinated with a primary course plus booster | 60 | 194 | 7.9 | -30.9-35.8 | 15 | -21.6- 41.0 |
| >80 years | | | | | | |
| Unvaccinated | 65 | 94 | | | | *reference* |
| Fully vaccinated with a primary course | 11 | 26 | 38.8 | -29.6-72.7 | 47.8 | -15.3-77.6 |
| Fully vaccinated with a primary course plus booster | 13 | 36 | 47.7 | -3.9-75.0 | 52.5 | 1.8-78.1 |

## Discussion

This study found that cases were generally older and had a higher prevalence of some of the comorbidities, including diabetes, neurological disease, and other underlying conditions, compared to controls. Interestingly, ICU admission and the need for oxygen supplementation were more frequent among controls, yet the mortality rate was significantly higher in the case group. The higher vaccination rates observed among controls suggest an association between vaccination and lower odds of SARS-CoV-2 positivity, as unvaccinated individuals were more commonly found among the cases. However, several subgroup analyses have wide confidence intervals crossing the null value, reflecting limited statistical precision. Therefore, these results should be interpreted as observational estimates rather than definitive evidence of significant protection.

### Vaccination and severe outcomes

Point estimates of VE were higher among fully vaccinated individuals particularly those with an additional booster, against severe outcomes among SARI patients. Despite the relatively low overall VE of 30.6% with just the primary course, this effectiveness increases to 37.8% with a booster dose, although confidence intervals were wide and overlapped zero Comparatively, other studies report higher VEs, but consistently show increased effectiveness with the administration of

booster doses. For instance, a study in Spain using data from 2021 reported VE estimates between 70% and 80% for fully vaccinated individuals, while a UK study conducted between December 8, 2020, and October 1, 2021, found that VE against hospitalization and death remained around 80% at 20 weeks post-vaccination. Similarly, an Israeli study analyzing data from July 30 through August 31, 2021 demonstrated that a third Pfizer-BioNTech dose significantly reduced rates of severe illness, particularly in older adults [13–15]. In addition, our study period coincided with Omicron-dominant circulation in Europe, during which VE against hospitalization has generally been lower than in earlier variant periods [7,8,16]. This aligns with comparator studies conducted during the Omicron wave between December 2021 and July 2022 Albania's use of a heterogeneous vaccine product mix, including viral vector and inactivated vaccines, together with the low vaccination coverage observed in this study population—where more than half of hospitalized SARI patients were unvaccinated and only around one fifth had received a booster dose (Table 1)—may also have contributed to the modest VE estimates.

**Impact of previous SARS-CoV-2 infection**

Patients with a prior history of SARS-CoV-2 infection had higher estimated VEwhen a booster was added to their primary vaccination course, with VE reaching 76.2%. This finding is consistent with observed benefitsof combining natural and vaccine-induced immunity [17] and further supports the role of booster doses in improving estimated protection particularly among previously infected individuals [18]. These findings align with results from the US Flu VE Network, which reported enhanced VE against outpatient illness during the Omicron variant's circulation using data up to August 8, 2021 [19]. Additionally, study from Israel analyzing data from August and September 2021 demonstrated that patients with hybrid immunity had reduced risks of reinfection and severe disease compared to those with only vaccination or natural infection, with both studies emphasizing that hybrid immunity provides broader and more durable protection, particularly against variants of concern [17].

From a policy perspective, these findings suggest that in settings with high levels of prior infection but low booster uptake, such as Albania [4], risk-based booster strategies targeting older adults and individuals with comorbidities may yield substantial public health benefit even when population-wide booster coverage remains limited. Consistent with this, WHO SAGE's 2023 guidance, synthesizing evidence collected since January 2021, emphasizes focusing booster efforts on high-risk groups (e.g., older adults and those with comorbidities) given widespread hybrid immunity in populations [5].

**Age-specific vaccine effectiveness**

The age-specific analysis revealed variations in VE. Patients over 80 years demonstrated the highest VE with booster doses, indicating that booster doses, suggesting greater estimated protection in older populations, who are generally more vulnerable to severe disease. Booster doses have been associated with enhanced immune protection in elderly populations, thereby reducing the risks associated with severe outcomes as demonstrated in studies using data from February through September 2021 and earlier periods up to mid-2021 [20,21]. In contrast, point estimates in the 60–79 age group were lower and confidence intervals were wide, limiting definitive interpretation in this age stratum. For patients aged 18–59 years, the VE estimates were 35.5% for the primary course and 42.8% after booster vaccination, compared with 15.5% and 15.0%, respectively, in the 60–79 group. This age group generally displayed a higher VE compared to the 60–79 group, likely due to fewer comorbidities and a stronger immune response. Booster doses among adults aged 18–49 in the United States, studied during December 2021, were associated with a reduced risk of hospitalization [22]. Furthermore, data spanning August 26, 2021, to January 22, 2022, indicated that while VE tends to decline over time post-booster, it still indicates meaningful estimated protection among younger adults [23]. Although VE in the 18–59 and 60–79 age groups is lower than in the 80+group, it remains consistent with protection, underscoring the different dynamics of immune responses across age groups and the role of booster doses in enhancing protection. Overall, our wide

confidence intervals across age groups necessitate caution, but the point estimates align with the biological expectation that booster doses are associated with greater protection.

### Limitations of the study

Despite these significant findings, our study has several limitations. First, as an observational study the test-negative design it is inherently susceptible to biases, including selection and information bias. Reliance on hospital-based SARI data may miss severe COVID-19 cases among patients who did not seek hospital care, introducing potential selection bias. Hospital length of stay was included in adjusted models to partially account for differences in clinical course between cases and controls, however this is an imperfect severity proxy and may be influenced by non-clinical factors (e.g., discharge/transfer practices, bed availability) and competing outcomes such as early death, so residual confounding may remain. Moreover, the exclusion of patients with incomplete vaccination records could influence the representativeness of our cohort. Vaccine product–specific analyses were not performed and therefore, VE could not be estimated by vaccine type or platform. The study did not include pediatric populations or patients below 18 years old, limiting the generalizability of the findings to younger age groups. Additionally, the study was limited by a lack of comprehensive genomic sequencing, which prevented stratification of vaccine effectiveness (VE) by SARS-CoV-2 variants or Omicron sub-variants. Omicron sub-variants differ in their degree of immune escape, with sub-variants such as BA.4 and BA.5 showing greater resistance to vaccine-induced antibodies [16]. Consequently, VE estimates in this study likely represent an average across multiple sub-variants, potentially attenuating overall effectiveness during periods dominated by more immune-evasive strains.

### Conclusion

In conclusion, this study provides real-world evidence from Albania highlighting the association between vaccination, particularly booster doses, lower odds of severe COVID-19 outcomes, especially among older adults. While the primary vaccination course was associated with moderate estimated protection, the addition of a booster was associated with higher vaccine effectiveness estimates. As previously described, our findings suggest that those with hybrid immunity—combining prior infection with vaccination—have higher estimated levels of protection. Age-specific analyses indicate that booster doses may be associated with greater effectiveness in the elderly, with higher estimated protection against severe disease in this vulnerable group. These results align with global studies and emphasize the critical role of booster vaccination strategies in maintaining robust immunity against severe COVID-19, especially in populations at higher risk. Beyond Albania, these findings contribute to evidence from settings with heterogeneous vaccine product mixes, moderate primary-series uptake, and limited booster coverage during Omicron-era transmission. In such contexts, modest point estimates of VE against severe outcomes may reflect a combination of immune evasion, waning protection since last dose, and high baseline population immunity from prior infection. The results therefore support risk-based booster strategies prioritizing older adults and individuals with comorbidities, and they highlight the value of strengthening routine severe respiratory infection surveillance platforms to generate timely, locally relevant VE estimates that can inform policy in resource-constrained and middle-income settings.

Further research is needed to explore the interplay between prior infection, vaccination, and age-related immune responses to optimize protection against evolving variants. Addressing the identified limitations through future research will be crucial in further understanding and enhancing COVID-19 vaccine effectiveness.

### Supporting information

**S1 Data. Anonymized dataset containing SARI COVID-19– positive cases and COVID-19–negative controls by week of test result, and vaccinated SARI cases by week of complete vaccination, Albania SARI sentinel surveillance, 1 July 2022–1 July 2023.**
(CSV)

## Acknowledgments

We are grateful to the sentinel general practitioners, paediatricians, virologists, and epidemiologists participating in the Albania SARI Surveillance System. We also extend our thanks to the WHO-Europe team, particularly Mark Katz and Iris Finci, for their valuable support in preparing this article. Special thanks to Nana Mebonia and Netta Beer for their contributions to this article as frontline coordinators in the MediPIET program, supported by ECDC.

## Author contributions

**Conceptualization:** Jonilda Sulo, Kujtim Mersini.

**Data curation:** Jonilda Sulo, Kujtim Mersini, Iris Hatibi, Najada Como, Esmeralda Meta, Migena Qato, Valbona Zefi, Juljana Nanaj.

**Formal analysis:** Jonilda Sulo, Kujtim Mersini.

**Investigation:** Jonilda Sulo, Elona Kureta, Najada Como, Esmeralda Meta, Migena Qato.

**Methodology:** Jonilda Sulo, Kujtim Mersini, Iris Hatibi, Eugena Tomini, Juljana Nanaj.

**Project administration:** Silva Bino.

**Software:** Jonilda Sulo, Valbona Zefi.

**Supervision:** Kujtim Mersini, Elona Kureta, Eugena Tomini, Silva Bino.

**Validation:** Jonilda Sulo, Kujtim Mersini, Eugena Tomini, Silva Bino.

**Visualization:** Jonilda Sulo.

**Writing – original draft:** Jonilda Sulo, Kujtim Mersini.

**Writing – review & editing:** Jonilda Sulo, Kujtim Mersini, Elona Kureta, Iris Hatibi, Najada Como, Esmeralda Meta, Migena Qato, Eugena Tomini, Silva Bino.

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
