## [Decision Letter · Decision Letter 0]

8 Jan 2026

PGPH-D-25-03497

COVID-19 Vaccine Effectiveness in Preventing Severe Outcomes and Assessing the Impact of Prior SARS-CoV-2 Infection Among Hospitalized Adults in Albania, July 2022-July 2023

Dear Dr. Sulo,

Thank you for submitting your manuscript to PLOS Global Public Health. After careful consideration, we feel that it has merit but does not fully meet PLOS Global Public Health’s publication criteria as it currently stands. Therefore, we invite you to submit a revised version of the manuscript that addresses the points raised during the review process.

Please respond to both reviewers' comments. I do note that there is complexity in how VE estimates are obtained and interpreted based on geography and timing - so your response to the reviewers on that point (why VE estimates vary compared to previous studies) does not need to be so extensive.

We look forward to receiving your revised manuscript.

Kind regards,

Abram L. Wagner, PhD, MPH

Academic Editor

Journal Requirements:

1. We note that your Data Availability Statement is currently as follows: Aggregated data supporting the findings are available within the article and its Supporting Information files. The underlying individual-level data are not publicly available because they originate from official national information systems and contain sensitive health information.

Reviewers' comments:

Reviewer's Responses to Questions

**Comments to the Author**

1. Does this manuscript meet PLOS Global Public Health’s publication criteria? Is the manuscript technically sound, and do the data support the conclusions? The manuscript must describe methodologically and ethically rigorous research with conclusions that are appropriately drawn based on the data presented.

Reviewer #1: Yes

Reviewer #2: Yes

2. Has the statistical analysis been performed appropriately and rigorously?

Reviewer #1: Yes

Reviewer #2: No

3. Have the authors made all data underlying the findings in their manuscript fully available (please refer to the Data Availability Statement at the start of the manuscript PDF file)?

Reviewer #1: No

Reviewer #2: No

4. Is the manuscript presented in an intelligible fashion and written in standard English?

Reviewer #1: Yes

Reviewer #2: Yes

5. Review Comments to the Author

Reviewer #1: The data for this study is not included in the submission. Even though the authors did mention that the underlying individual-level data are not publicly available because they originate from official national information systems and contain sensitive health information.

The manuscript is well written. It evaluates COVID-19 vaccine effectiveness (VE) in preventing severe outcomes among hospitalized patients in Albania using a test-negative case-control design using the SARI sentinel surveillance data (July 2022–July 2023). The study is relevant, it does address a gap in vaccine effectiveness data in Albania and the Balkan region generally. The findings also highlight a modest VE for primary and booster vaccinations, with substantially higher protection when combined with prior infection, especially among elderly patients. The work potentially contributes some evidence informing vaccination strategies in less developed countries.

Minor Comments

- Abstract: The phrase “VE against severe outcomes was 30.6%, increasing to 37.8% with a booster” could be misinterpreted; could the authors clarify if these are adjusted estimates? Starting the sentence in the 'results section' with the full meaning of VE is more acceptable.

- Introduction: Can the authors consider an Albania-specific background after a short intro of the global COVID-19 background?

- References: Can the authors kindly update with the latest WHO SAGE recommendations on hybrid immunity and booster strategies?

Major Comments

1. METHODS:

- In the description of inclusion/exclusion criteria, Fig 1 is mentioned but not fully described in the text.

- Can the authors kindly clarify how the documented co-morbidities were confirmed (through self-report or medical records, or any other means)?

2. VE ESTIMATES:

- Some confidence intervals appear wide & with negative values, which presupposes a limited sample size and/or statistical uncertainty, can the authors clarify.

- There appears to be a relatively low VE compared to other European studies, can the authors comment on possible reasons e.g. due to vaccine types used, coverage rates, Omicron predominance, etc).

3. A Priori Infection Analysis:

- The authors describe an increase in VE when combined with prior infection (76.2%). Please could the authors expand on this in the discussion about hybrid immunity and perhaps policy implications, especially in settings with low booster uptake.

4. Limitations

- The manuscript acknowledges limited viral typing capacity. Could the authors expand on how this may affect VE estimates, given Omicron sub-variant diversity in the limitations section?

- Can the authors describe potential biases from hospital stay duration as a proxy for severity?

In summary, the authors can strengthening the discussion of VE interpretation, clarifying the methodological details, and expand on hybrid immunity implications to improve impact and relevance of this manuscript.

Thank you for this write-up.

Reviewer #2: Dear Authors,

This study provides important regional data on COVID-19 vaccine effectiveness (VE) in Albania, filling a notable geographic gap in public health literature. However, to meet the rigorous standards for publication, a few methodological and interpretive issues must be addressed.

Major Issues

1. Statistical Precision and Confidence Intervals: A primary concern is the statistical power of the study. Many VE estimates feature very wide confidence intervals (CIs) that cross the null value or include negative values. In clinical and public health research, when a confidence interval for VE crosses zero, the result is not considered statistically significant, meaning the study cannot definitively conclude that the vaccine provided protection in that specific subgroup. The authors should explicitly acknowledge this lack of significance in the discussion sections. Presenting these point estimates as definitive "protective effects" without emphasizing the uncertainty (the wide CIs) is misleading.

2. Discrepancy in VE Values: The reported VE of 30–38% is considerably lower than the 70–90% often reported in other European studies during similar periods. While the authors mention rollout timing, a deeper analysis is required regarding Vaccine Product Mix and Variant Dynamics:

Minor Issues

1. Table clarity and data presentation

• Table 1 Formatting: The "Age" row currently presents "53-74" and "66" in a single cell, which is highly confusing. This must be reformatted to clearly distinguish between the Median and the Interquartile Range (IQR).

• Inconsistency: The number of decimal points varies throughout the tables. For professional reporting, please standardize to one or two decimal places consistently.

2. Over-interpretation of results: The authors describe VE estimates of 30–38% as "substantial." In the context of global COVID-19 research, where "substantial" is typically reserved for effectiveness above 60–70%, this characterization is an over-interpretation. A more neutral term would be more scientifically accurate given the data provided.

3. Global Public Health Framing: To increase the manuscript's impact, the authors should broaden the discussion beyond the Albanian context.

6. PLOS authors have the option to publish the peer review history of their article (what does this mean?). If published, this will include your full peer review and any attached files.

**Do you want your identity to be public for this peer review?** For information about this choice, including consent withdrawal, please see our Privacy Policy.

Reviewer #1: No

Reviewer #2: No

Figure Resubmissions:

---

## [Editor Report · Decision Letter 1]

2 Mar 2026

PGPH-D-25-03497R1

COVID-19 Vaccine Effectiveness in Preventing Severe Outcomes and Assessing the Impact of Prior SARS-CoV-2 Infection Among Hospitalized Adults in Albania, July 2022-July 2023

Dear Dr. Sulo,

Thank you for submitting your manuscript to PLOS Global Public Health.  I appreciate your response to reviewers, and I just have a few comments to add before I can make a decision.

We look forward to receiving your revised manuscript.

Kind regards,

Abram L. Wagner, PhD, MPH

Academic Editor

Journal Requirements:

Additional Editor Comments (if provided):

Major Comments

1. Exclusion of influenza-positive cases (line 115)

You mention excluding influenza-positive cases from the analysis. Could you please provide additional justification for this decision? Ideally, this would include either:

A citation supporting this analytic approach, or

A sensitivity analysis in which influenza-positive cases are included (e.g., as negative controls) to assess how this impacts your primary results.

Given that many of your estimates are non-significant, I would not expect large shifts, but including such an analysis would strengthen the methodological rigor and reassure readers that findings are robust to this specification.

2. Contextualization of COVID-19 burden in Albania (lines 41 and related discussion)

I appreciate that you have added additional information about the epidemic in Albania, as requested by prior reviewers. However, I think further contextualization would strengthen the manuscript.

You note 320,000 confirmed cases in 2022–2023 (line 41). It would be helpful to situate this in context. For example:

What proportion of the population does this represent?

How does this compare with estimates of total population exposure (as opposed to confirmed cases)?

Given that vaccine effectiveness is strongly influenced by prior infection and hybrid immunity, as your results suggest, it would be useful to briefly discuss the difference between confirmed cases and likely cumulative infection burden. If available, are there any seroprevalence studies from Albania or neighboring countries that could provide insight into the proportion of the population with prior immunity at the time of your analysis? Even one or two sentences acknowledging this distinction would improve the interpretation of your findings.

3. Timing of comparator studies (Discussion section)

When comparing your vaccine effectiveness estimates to other studies, could you please specify the dates during which those studies were conducted (not just publication dates)? Vaccine effectiveness varies substantially by circulating variant and calendar time, so indicating when comparator studies took place would help readers better contextualize similarities and differences.

4. Clarification of “variants not included” (line 90)

In the Methods, you state that variants were “not included.” Could you clarify what this means operationally? For example:

Were variant data unavailable and thus all cases were grouped together?

Were variant-specific cases categorized as COVID-positive without differentiation?

Were variant data treated as missing?

5. Analysis of severe outcomes

For your vaccine effectiveness analysis of severe outcomes, could you provide more detail on how these models were structured? How were non-severe cases handled in that analysis (excluded, treated as controls?)?

Minor Comments

1. Use of causal language (Abstract conclusion and elsewhere)

In the abstract and elsewhere, you use the word “effect.” Because this is an observational study, I recommend avoiding causal language. Unless referring specifically to “vaccine effectiveness” as a standard epidemiologic term, I suggest replacing “effect” with “association” or similar non-causal terminology throughout the manuscript.

2. Clarification of Comirnaty (lines 50–57) - clarify if this is a brand name, manufacturer, or product designation.

3. Table 1 – Mean vs. median

In Table 1, you list a mean but later describe median and interquartile range (IQR) in your response to reviewers.

4. Table 2 – Confidence interval formatting

In Table 2 (third row), part of a confidence interval appears to be missing.

5. Language comparing vaccine effectiveness estimates (line 214 and following paragraph)

When comparing vaccine effectiveness estimates (e.g., primary course 30.1% vs. booster 31.4%), I suggest avoiding language such as “higher” or “lower” when the estimates are numerically similar and overlapping. Simply presenting the estimates side-by-side without comparative wording would be clearer and more neutral. The same applies to the following paragraph.

6. Formatting of large numbers

Please ensure consistency in formatting numbers with four or more digits (e.g., inclusion or exclusion of commas in the thousands place). Choose one format and apply it consistently throughout.

Reviewers' comments:

Figure Resubmissions:

---

## [Editor Report · Decision Letter 2]

19 Apr 2026

COVID-19 Vaccine Effectiveness in Preventing Severe Outcomes and Assessing the Impact of Prior SARS-CoV-2 Infection Among Hospitalized Adults in Albania, July 2022-July 2023

PGPH-D-25-03497R2

Dear Msc Sulo,

We are pleased to inform you that your manuscript 'COVID-19 Vaccine Effectiveness in Preventing Severe Outcomes and Assessing the Impact of Prior SARS-CoV-2 Infection Among Hospitalized Adults in Albania, July 2022-July 2023' has been provisionally accepted for publication in PLOS Global Public Health.

Best regards,

Abram L. Wagner, PhD, MPH

Academic Editor